# Epstein–Barr Virus Promotes B Cell Lymphomas by Manipulating the Host Epigenetic Machinery

**DOI:** 10.3390/cancers12103037

**Published:** 2020-10-19

**Authors:** Andrea Di Pietro

**Affiliations:** Infection and Immunity Program and The Department of Biochemistry and Molecular Biology, Biomedicine Discovery Institute, Monash University, Clayton, VIC 3800, Australia; andrea.dipietro@monash.edu

**Keywords:** EBV, epigenetics, chronic inflammation, B cells

## Abstract

**Simple Summary:**

Epstein-Barr Virus (EBV)-induced lymphomas have a significant global incidence, given the widespread infection to the human population. EBV adopts several mechanisms to replicate and persist in the host, by hijacking its epigenetic machinery. The main topic of this review details the current insights of EBV interactions with the host epigenetic system, and it will be discussed the potential relationship between the EBV-induced chronic inflammation and the dysregulation of epigenetic modifiers that might lead to tumorigenesis. Promising novel therapies against several types of cancer involve the use of epigenetic modifier inhibitors. To design new therapeutical strategies targeting lymphomas, it is crucial to conduct exhaustive reaserch on the regulation of these enzymes.

**Abstract:**

During the past decade, the rapid development of high-throughput next-generation sequencing technologies has significantly reinforced our understanding of the role of epigenetics in health and disease. Altered functions of epigenetic modifiers lead to the disruption of the host epigenome, ultimately inducing carcinogenesis and disease progression. Epstein–Barr virus (EBV) is an endemic herpesvirus that is associated with several malignant tumours, including B-cell related lymphomas. In EBV-infected cells, the epigenomic landscape is extensively reshaped by viral oncoproteins, which directly interact with epigenetic modifiers and modulate their function. This process is fundamental for the EBV life cycle, particularly for the establishment and maintenance of latency in B cells; however, the alteration of the host epigenetic machinery also contributes to the dysregulated expression of several cellular genes, including tumour suppressor genes, which can drive lymphoma development. This review outlines the molecular mechanisms underlying the epigenetic manipulation induced by EBV that lead to transformed B cells, as well as novel therapeutic interventions to target EBV-associated B-cell lymphomas.

## 1. Introduction

Epigenetic modifications are the foundation of plasticity in multicellular organisms, which are characterised by functionally distinct cell types while sharing identical genome sequences. The epigenetic machinery induces and sustains unique gene transcriptional patterns that determine the developmental fate of each cell type [1]. There are three broad types of epigenetic mechanisms within cells that regulate chromatin accessibility and gene expression: DNA methylation, histone modification, and posttranscriptional gene regulation by non-coding RNA (miRNA) [2]. Each of these mechanisms plays a vital role in a wide variety of essential biological processes, ranging from organismal development and cell differentiation to cellular stress responses, tissue homeostasis, and formation of immune memory [3,4].

Recent advances in the epigenetic field have revealed that the perturbation of the epigenetic landscape leads to alterations in gene expression, ultimately resulting in cellular transformation and malignant outgrowth [5]. While the epigenetic machinery is finely regulated in normal cells, there is growing evidence indicating that environmental factors can induce epigenetic alterations by affecting the expression and functionality of enzymes involved in DNA methylation or histone modifications. The breakdown of the epigenetic mechanisms causes variations in transcriptional programs of cells and might eventually lead to tumorigenesis. Among these factors, several oncogenic and persistent DNA and RNA viruses contribute to epigenetic changes characteristic of cancerous cells [6], which include the Epstein–Barr virus (EBV) [7], Hepatitis viruses (HBV, HCV) [8], Kaposi’s sarcoma herpesvirus (KSHV) [9], and numerous human papillomaviruses (HPVs) [10].

EBV, also known as human herpesvirus 4 (HHV4), is a ubiquitous gammaherpesvirus which asymptomatically infects >95% of the world population. The primary infection occurs in oropharyngeal epithelial cells [11]; however, EBV predominantly infects B lymphocytes during its latent stages. Within the immunocompetent individual, the virus persists in naïve and memory B cells in a non-pathogenic state for the lifetime of the host [12]. Intermittently, these virus-infected memory B cells differentiate into antibody-secreting plasma cells, which awakes EBV from its latent stage, leading to infection of other resting B lymphocytes [13]. In immunocompromised hosts, such as postoperative organ transplant or HIV-infected patients, EBV infection is strongly associated with several B-cell lymphomas. The list includes endemic/sporadic Burkitt’s lymphoma (eBL/sBL), diffuse large B-cell lymphoma (DLBCL), classical Hodgkin’s lymphoma (cHL), primary central nervous system lymphoma (PCNSL), primary effusion lymphoma (PEL), and plasmablastic lymphoma (reviewed in [14]).

Epigenetic modifications play a critical role in EBV-associated B-cell lymphoma development. In particular, EBV viral products themselves mimic the epigenetic modifiers of the host by directly manipulating gene expression profiles [15]. One of the most critical viral factors is the nuclear antigen EBNA2, which induces B cell activation and proliferation [16]. EBNA2, together with EBNA-LP (leader protein), co-regulates the expression of several host genes, including the proto-oncogene MYC and transcription factors, such as PU.1, EBF1, IRF4, and CBF1 [17]. During the latent stages, additional EBV viral products such as latent membrane proteins (LMPs) and non-coding RNAs (ncRNAs) transform B cells into immortalised lymphoblastoid cell lines (LCL) in vitro and induce memory cell differentiation in vivo, by silencing the tumour suppressor *PRDM1* [18]. The direct role of EBV viral products in manipulating the epigenetic landscape of infected B cells has been explored over the past decade. This review will provide a comprehensive overview of the well-known mechanisms behind this manipulation and the latest advances in the field.

Studies on host–pathogen interactions have illuminated a role for the inflamed environment generated by infectious pathogens in altering the functionality and expression of epigenetic modifiers. Given the strong link between inflammation and tumorigenesis, but also epigenetic dysregulation and cell transformation, it would be crucial for new therapeutic strategies to explore how the pathogen-mediated inflammation could modulate the epigenetic landscape in the context of cancer. A noteworthy review by P. Yang et al. discussed the association between Hepatitis B virus (HBV)-related inflammation and the initiation of hepatocellular carcinoma (HCC) in chronically diseased liver tissue [19]. A model has also been proposed to link the HBV-inflammation environment and the epigenetic deregulation during HCC development and progression, generating a new working hypothesis on how other chronic infections may alter the epigenome to induce tumorigenesis [20].

Little is known about the link between the inflammation status caused by EBV infection and the regulation of epigenetic modifiers. For this reason, in the second part of this review, we will discuss the role of the inflammatory environment mediated by EBV and the causative factors regulating the epigenetic landscape, leading to B cell transformation.

## 2. Host–Pathogen Interaction during EBV Pathogenesis

EBV has a complex life cycle, with distinct cellular tropism depending on the stage of infection. Dissecting each phase of the host–pathogen interaction is critical to understand the potential of EBV to transform B cells (Figure 1) [21].

### 2.1. The Viral Life Cycle: From the Epithelium to B Cells

EBV is typically transmitted by saliva and its initial infection, known as the lytic replication cycle, affects the oropharyngeal epithelial cells in the upper respiratory tract. In children, it is usually asymptomatic; however, the primary infection in adults can be more severe, leading to a syndrome known as infectious mononucleosis (IM) [22]. During the lytic cycle, EBV crosses the mucosal barrier, and it spreads into the bloodstream. Here, it targets circulating naïve and memory B cells, through the binding of the major envelope glycoprotein gp350/220 to the CD21 receptor that is highly expressed on these populations [23]. Both B cell populations represent the reservoir for the establishment of EBV latency. Unlike the lytic cycle, latency does not result in virion production; nevertheless, during the latent phase, subsets of EBV genes are expressed following four distinct latency programs: latency 0, I, II or III. Intermittently, the virus can reactivate and re-enter the lytic replication cycle. However, the host immune system is sufficient to maintain control of the infection, by inducing a robust cytotoxic T lymphocytes (CTL) response [24] and producing neutralising antigen-specific IgG antibodies (Figure 2) [25]. In immunocompromised individuals, the immune system fails to monitor viral replication, resulting in an increase in latently infected cells in the peripheral blood or persistently infected cells on the oropharynx. Similarly, patients co-infected with other chronic pathogens (HIV, Plasmodium Falciparum) maintain a persistent status of B cell activation and memory formation [26,27,28], which may contribute to an expanded number of latently EBV-infected, proliferating B cells. In both scenarios, EBV latency programs can lead to malignant transformation of B cells.

### 2.2. EBV Adopts Immune Evasion Strategies to Facilitate Viral Persistence in B Cells

EBV expresses differential sets of proteins depending on the stage of infection, giving the virus a survival advantage by evading the immune response and facilitating viral persistence in B cells. During the lytic replication, the early-immediate BZLF1 and BRLF1 molecules act as transactivators that induce the expression of ~30 early lytic genes but, on the other hand, interfere with antiviral signals by inhibiting the interferon regulator factor 7 (IRF7) transcriptional activity [29] and reducing the expression of TNF*α*, IFN*γ* and HLAI/II [30,31,32]. Despite the modulation of antiviral functions, the lytic stage of infection still induces a robust cellular and humoral response towards the different viral proteins, by recruiting both B and T cells at the site of primary infection. However, EBV can quickly modulate antigen-presenting pathways by reducing both HLAI and II molecules in target cells [30]. The decreased expression of HLA molecules slows the adaptive response against the pathogen, which in turn takes advantage of this temporal delay to infect naïve and memory B cells, where it eventually establishes latency. Additionally, the virus adopts similar immune evasion strategies in the newly infected B cells by activating the expression of BGLF5, another viral protein with immune-evasive properties. Indeed, BGLF5 is directly involved in the degradation of HLAI/II mRNAs [33]. This additional inhibition of cross-presentation functions makes EBV+B cells poorly recognisable by CD4 or CD8 T cells for their cytotoxic activities [34]. The fine modulation of these signalling pathways leads to diminished adaptive responses due to a lower interferon responsiveness and hampered antigen presentation, allowing the establishment of infection and persistence.

### 2.3. Targeting B Cells to Establish Viral Persistence

During the latent stages, EBV switches to different gene expression programs compared to the lytic phase. These latent proteins help in maintaining the persistence of the pathogen but also inducing malignant growth in immunocompromised patients. Mechanisms adopted by EBV during the latent stages include manipulation of both cellular signalling pathways and epigenetic machinery. There are two proposed pathways to latency establishment; (i) following direct infection of memory B cells or (ii) via a germinal center (GC) dependent process in which naïve B cells infected with EBV traverse through GC reactions and emerge as memory cells harbouring the virus [35]. Several latent programs induced by EBV that finely regulate the persistency stages have been described, which are distinct and sequentially regulated depending on the target cell (Figure 1). Each of these programs is dictated by the expression of distinct EBV proteins, which are finely regulated by the activation or repression of viral latent promoters (Cp, Wp, Qp, and LMPp). Latency III, or the Growth Program, is restricted to lymphoblasts following the expression of Epstein–Barr nuclear antigens (EBNA1, -2, -3, -4, -5, -6) and latent membrane proteins (LMP1, LMP-2A, -2B) [36]. Latency II, also called the Default Program, is established in infected germinal centre centroblasts and includes the expression of EBNA-1 and LMP1, LMP2A, -2B. The selective pressure driven by CTLs leads the transition to the stricter latency program II, likely giving EBV a survival advantage. This is accounted for the increase in DNA methylation of the Wp and Cp, silencing the expression of the highly immunogenic EBNA proteins [37,38]. Latency 0 and I (Latency Programs) are restricted to resting memory B cells and are characterised by the lack of any viral gene expression, or in the latter only the expression of the weakly immunogenic EBNA-1. During this phase, EBV epigenome is fully methylated except for the promoter Qp, which drives the expression of EBNA-1, responsible for latency maintenance [39].

### 2.4. The Reactivation of EBV Lytic Replication Increases the Pool of Infected B Cells in Immunocompromised Patients

EBV remains in the latent state for most of the host’s lifetime; however, physiological stimuli can reinduce the lytic infection. The impaired immunosurveillance of immunocompromised patients fails in monitoring the secondary infection, which eventually causes an increase in newly EBV+ B cells, increasing the risk of B cell transformation [40]. The causative factors behind this viral awakening can be differentiated into two primary sources: (i) cellular signalling pathways, including B-cell receptor (BCR) antigen stimulation and downstream signals, and (ii) cellular stresses induced by external factors, such as oxidative stress, hypoxia, and inflammation. Among these, it is becoming clear that the pro-inflammatory state caused by other chronic co-infections is a crucial feature that disturbs the latent state of the virus.

For example, *P. Falciparum* malaria is known to interfere with both EBV biology and EBV-specific immunity, leading to EBV reactivation from its latent state. During the course of malaria, the *P. falciparum* erythrocyte membrane protein 1 (PfEMP1) adheres to and activates B cells through the cysteine-rich interdomain region 1α (CIDR1α). This interaction triggers the expression of Toll-like receptor 7 and 10 (TLR7 and -10) on B cells, inducing a persistent activation of B cells [41]. The pro-inflammatory signals induced by malaria parasites have been shown to concomitantly induce EBV reactivation in EBV+B cells [42]. Indeed, a clinical study showed that holoendemic malaria results in elevated EBV viral loads, and this is eventually associated with higher impact of the endemic Burkitt Lymphoma (eBL) in sub-Saharan regions [43]. Following the same principles, chronically infected and untreated HIV+ patients have a higher probability of developing EBV-associated malignancies (reviewed in [44]). HIV has been long known to hyperactivate B cells, with terminal differentiation into plasmablasts and plasma cells inducing EBV reactivation and contributing to the increase of its reservoir. However, the factors contributing to B cell hyperactivation and expansion of EBV-infected B cells remain largely unknown.

## 3. Role of Epigenetics in EBV Infection and Cancer Formation

One of the primary mechanisms by which EBV distinctly regulates its transcriptional programs lies in the ability of this pathogen to manipulate the epigenetic machinery of the host. Given that several cancers feature epigenetic alterations, it is not surprising that EBV can lead to B cell transformation.

### 3.1. Latent Proteins Modulate the Host Epigenetic Machinery for Viral Silencing

During latency establishment, the EBV genome undergoes chromatin reorganisation; whereby it circularises into episomes by recombination of its terminal repeats, it assembles into nucleosomes, and methyl groups are deposited on GpG islands [45]. The pre-latent stage is characterised by the specific repression of the early transactivator genes BZLF1 and BRLF1 by EZH2, the core subunit of the polycomb repressive complex 2 (PRC2), which is responsible for the deposition of methyl groups on lysine 27 on the histone H3 tail (H3K27me3) [46,47,48]. EBV latent proteins, such as EBNA1 and -2, can directly induce genetic alterations or recruit several chromatin remodelling factors. For example, EBNA1 can interfere with the function of the maintenance DNA methyltransferase (DNMT1) [49] and recruit the histone deubiquitylase USP7 [50], resulting in the demethylation to the *OriP,* the latent origin of EBV replication. LMP1 and LMP2A act as direct epigenetic modifiers by directly inducing the expression and activity of DNMT1, 3A and 3B [51,52,53], and the H3K27 demethylase KDM6B [54].

EBV has evolved to persist in the host by interacting with and manipulating the functions of several epigenetic modifiers. This peculiarity provides this pathogen with the potential to disrupt the epigenetic landscape of the infected cells, staging the basis of cancer.

### 3.2. Molecular Basis of B Cell Transformation

The deterioration of host immune surveillance against EBV, followed by an increased number of newly infected B cells and the re-establishment of latent programs, are the initial steps preceding cell transformation. EBV-mediated B-cell immortalisation is associated with the global alteration of both epigenetic landscape and cell gene expression [55]. Among the EBV viral products, five viral latent antigens, EBNA2, EBNALP, EBNA3A, EBNA3C, and LMP1, are essential for efficient B cell transformation [56] (Table 1).

#### 3.2.1. EBNA2, EBNALP

EBNA2 is mostly expressed during the latent III state and has a crucial role in B cell transformation found in diffuse large B cell lymphomas (DLBCLs). One of the most studied mechanisms in B cell transformation is the EBNA2-mediated activation of the MYC proto-oncogene, which promotes cellular growth [17]. EBNA2 binds at multiple upstream and downstream enhancers and super-enhancers (SE) in the MYC promoter region. EBNA2 recruits the BRG1 ATPase of the SWI/SNF chromatin remodeling complex to MYC-targeting enhancers, and organises the DNA loops along a 3 Mb region of the MYC promoter [57]. TET2, a cellular tumour suppressor involved in active DNA demethylation, plays a central role in regulating the DNA methylation state during EBV latency. EBNA2 has a dual role in the regulation of TET2: (i) it activates TET2 expression by directly binding to the EBNA2-dependent RBP-jκ and EBF1 binding sites in TET2 promoter [58,59], and (ii) it physically interacts with TET2 protein to cooperatively demethylate genes essential for EBV-driven B-cell growth transformation [58]. EBNALP cooperates with EBNA2 to activate viral and cellular gene transcription through removing repressive complexes from promoters, enhancers, and matrix-associated deacetylase bodies [60].

#### 3.2.2. EBNA3 Family Proteins

Genetic studies revealed that EBNA3A and EBNA3C are necessary for B cell transformation, whereas EBNA3B seems to be dispensable [61]. EBNA3A and EBNA3C cooperatively act as predominant viral oncoproteins through regulating cellular gene transcription. In particular, these two viral products activate the expression of BAFT and IRF4 [62], which are both critical for the EBV repressive function of the BCL2L11 gene that encodes for the pro-apoptotic factor BIM. It has been proposed that the BAFT/IRF4 complex can dock EBNA3A/C and other subunits of the polycomb repressive complex to actively silence BCL2L11 [63]. Moreover, EBNA3A/C interacts with the transcriptional repressors CtBP1 and 2, and target the cyclin-dependent kinase inhibitor 2A (CDKN2A, or p16^INK4a^); this complex represses the expression of this tumour suppressor gene and eventually leads to the outgrowth of EBV+ cells [64,65].

#### 3.2.3. Latent Membrane Proteins

Among the various latent membrane proteins, LMP1 and LMP2A are the major factors involved in tumorigenesis processes due to their mimicry feature of the CD40 and B cell receptor signalling pathway [66]. Besides CD40 signalling, LMP1 works in tandem with the DNA methyltransferase DNMT3B, regulating cellular apoptosis through activation of the NF-*κ*B pathway by elevating antiapoptotic Bcl2 expression [67]. LMP1 also induces DNMT1, which is involved in the JNK-AP-1 pathway [68]. LMP2A activates DNMT1 through the STAT3 signalling pathway and increasing IL-10 production to promote cell survival [69].

Overall, EBV encodes the capacity to manipulate the host epigenetic machinery through several viral effectors. The uncontrolled EBV replication and the modulation of the epigenetic landscape eventually lead to the development of lymphomas.

## 4. Inflammation, Epigenetics, and Tumorigenesis

A secondary aspect, though equally important, to be taken into account during B cell transformation is that EBV infection, especially during lytic reactivation, induces strong inflammatory responses. Studies on other chronic infection, such as HBV [70,71] and *Helicobacter Pilory* [72], have suggested that persistent inflammatory environments can be causative of epigenetic dysregulation and could ignite tumorigenesis processes. The main events that induce chronic inflammation during EBV pathogenesis might interfere with the epigenetic regulation of B cells, which eventually promotes neoplastic lesions. Given that EBV secondary infection is not controlled in patients with impaired immune responses, understanding the molecular aspects that drive the inflammatory response and its persistence is crucial in the design of new therapeutic strategies to prevent the induction of B cell transformation by this virus.

### 4.1. How Can Chronic Inflammation Drive Cancer Development?

The immune system plays a distinct role during tumour initiation, promotion, and progression, which is often referred to as “cancer-promoting inflammation”. Chronic inflammation, which usually occurs during persisting infections or chronic inflammatory diseases, has a significant impact on the composition of the tumour microenvironment (TME) [73]. Indeed, while the inflammatory environment is reduced to normal levels following a typical response to a pathogen or tissue damage, a persistent chronic inflammatory reaction might induce tumorigenesis through several mechanisms. Chronic inflammatory diseases are characterised by epigenetic changes, including altered histone modifications, DNA methylation, and ncRNA expression. While there is limited direct evidence, it is hypothesised that inflammation induces these epigenetic alterations that then contribute to cell transformation. One of the proposed mechanisms for this is that the persistent source of inflammation induces high levels of reactive oxygen species (ROS) [74], which in turn affects the expression and functionality of various epigenetic modifiers [75].

Infections with certain bacteria, viruses, or parasites have been associated with epigenetic alterations, DNA damage, and the development of cancer. For example, infection with *Helicobacter Pylori* is associated with chronic inflammation of the stomach (gastritis) and increased risk of gastric cancer and mucosa-associated lymphoid tissue (MALT) lymphoma [76]. The persisting inflammation caused by this bacterium is associated with a dysregulation of the epigenetic landscape, including genome-wide DNA hypomethylation [77], global de-phosphorylation of the histone residue H3Ser10, and deacetylation of the histone residue H3K23 [78]. Hepatitis B and C viruses are well-known etiological agents for hepatitis, and they are associated with increased risk of cirrhosis and hepatocellular carcinoma. Hepatitis viral infection causes a high level of ROS-induced DNA base damage [79], increasing the DNA methylation at the promoter of tumour suppressor genes including Cyclin-dependent kinase inhibitor 2A (CDKN2A, or p16) [80], E-cadherin (CDH1) [81], and Insulin-like growth factor binding protein 1 (IGFBP-1) [82]. The persistent infection also induces the tri-methylation of the histone residue H3K4, driving the expression of the oncogene Myc [83].

It has been hypothesised that *P. falciparum* in holoendemic areas is likely carcinogenic in humans, given the higher incidence of Burkitt Lymphoma among infected individuals [43]. The series of events that leads to lymphoma formation could be through the reactivation of EBV latency, which bears the oncogenic potential, or through the direct dysregulation of the host immune response. GC B cells isolated from tonsils in patients chronically infected with *P. falciparum* or from uninfected individuals, showed how this population was primarily amplified in chronically infected individuals. The majority of these cells were latently infected with EBV and displayed overexpression of activation-induced cytidine deaminase (AID) [84], suggesting an indirect role of malaria-mediated inflammation in harbouring a suitable environment for EBV latency and increased risk of developing endemic Burkitt Lymphoma [85].

Together, these examples show that alterations in the epigenetic landscape are initiated and driven by the chronically inflamed environment established by specific pathogenic agents, which can eventually lead to tumour formation and development.

### 4.2. Cross-Talk between EBV-Mediated Inflammation and Epigenetic Regulation

As described in Section 2.2, EBV has adopted several strategies during its evolution to recall but evade the host immune system in order to establish the different phases of infection without killing the host. Indeed, the peculiarity of this virus is that, while it can dampen antiviral defences, it maintains high levels of inflammation.

During the lytic cycle or reactivation, EBV is recognised by the different TLRs expressed on both epithelial cells and lymphocytes, and downstream signalling through these receptors induces the production of type I interferon and other pro-inflammatory cytokines. Among them, TLR9 is the major TLR expressed in B cells and is responsible for the induction of IFNs, IL-6, and TNF-*α* [86]. However, the lytic protein BGLF5 targets TLR9 mRNA for degradation in infected B cells, inhibiting type I IFN production [87]. This mechanism is also maintained during latency, when LMP1 targets TLR9 by inhibiting its promoter activity [88]. On the other hand, LMP1 is also able to interact with TRAF proteins [89], which are identified as adaptor proteins in TLR signalling, thereby activating downstream NF-*κ*B signalling to promote cell growth and survival. While EBV effectively suppresses the antiviral effect of TLR signalling, the pro-inflammatory response triggered by TLR signalling is still prominent under some circumstances, and it remains to be elucidated how these signals can regulate the expression of epigenetic modifiers. It has been demonstrated that the function of EZH2 (PRC2) is involved in the direct repression of EBV lytic genes BZLF1 and BRLF1 for the initiation of the latent phases [46,47,48], but it is not clear how this subunit is regulated during EBV pathogenesis. A few selected studies have shown that chronic activation of NF-kB can induce EZH2 expression in CD40L-stimulated cells from Chronic Lymphocytic Leukemia patients [90]. This pathway might provide a mechanism through which microenvironment induced NF-kB can inhibit tumour suppressor functions and promote tumorigenesis.

When expressed in cells, IFNs execute their antiviral functions via the JAK-STAT signalling pathway, leading to the expression of interferon stimulated genes (ISGs). It is, therefore, not surprising that EBV has developed multiple strategies to counteract not only IFN production but also IFN signalling. LMP1 has an N-terminal transmembrane domain, which directly interacts with TYK2 kinase involved in type I and type III IFN signalling and suppresses phosphorylation of both STAT1 and STAT2 [91]. LMP1 can then block IFN-mediated antiviral responses in infected cells. However, in addition to STAT1 and STAT2, other STATs also play roles in cytokine signalling. Again, LMP1 is able to interact with and activate STAT3 without the typical induction of the cytokine IL-6 [92]. This interaction increases the production of other pro-inflammatory cytokines (such as IL-17), but most importantly, induces cellular growth and survival. Notably, cancer genome landscape studies have implicated mutations and dysregulation in various epigenetic modifiers as well as the JAK–STAT pathway as underlying causes of many cancers, particularly acute leukemia and lymphomas. It has been recently reported that STAT3 can bind to the promoter region of the EZH2 gene in gastric cancer cells, implicating STAT3 as a direct regulator of EZH2 [93]. STAT3 also mediates oncogenesis by recruiting DNA methyltransferase 1 (DNMT1) to gene promoters to silence tumours suppressor genes, such as PTPN6, IL-2Rγ, CDKN2A, DLEC1, and STAT1 by CpG methylation in malignant T lymphocytes and breast cancer cells [94,95]. It has also been shown that STAT5 can recruit the DNA demethylases TET2 [96], which is also involved in the establishment of EBV latency. In light of these findings, it is crucial to understand the interplay between EBV-mediated inflammation, STAT proteins, and regulation of chromatin remodelling factors and how these can various aspects interact to promote cancer.

## 5. Open Questions

We have described here how the uncontrolled immune response to EBV, triggered by co-infection or other exacerbating diseases, leads to B cell malignancies that arise due to dysregulation of the epigenetic landscape in B cells. We have also discussed how chronic inflammation mediated by other persistent infections alters the expression and function of epigenetic modifiers, leading to tumorigenesis and cell transformation. While there is an understanding of how EBV viral products mimic the functions of epigenetic modifiers for lymphoma development, there is a major gap in knowledge regarding the dysregulation of these molecules driven by the EBV-mediated inflammatory state. Which specific cytokines are involved in the regulation of epigenetic modifiers in the context of EBV infection? Which epigenetic modifiers specifically target tumour-related genes? Exploring the biology behind the direct link between the EBV-mediated inflammation and epigenetics represents an exciting field of research, which may lead to the design of new therapeutic strategies to fight B cell lymphomas.

## 6. Conclusions and Future Perspectives

Over the past two decades, the field of epigenetic regulation and functions has received great attention due to the intricate interplay between viruses, cellular transcription factors, and histone-modifying enzymes. Importantly, it has been revealed that mutations and dysregulation of epigenetic modifiers lead to several types of cancers, due to their transforming ability in inducing cell survival and proliferation. Numerous oncogenic viruses, such as EBV and hepatitis viruses, have a direct role in manipulating the expression of these molecules, and their particular viral persistent feature has the potential to transform target cells and the infected tissue. Many viral products bear the ability to mimic or recruit epigenetic modifiers to cellular target genes for the establishment and maintenance of distinct infectious stages, including latency. Moreover, in response to the host immune system, these viruses have evolved evasion strategies by hijacking various cellular regulatory mechanisms. As discussed in this review, the modulation of the host epigenetic machinery represents a crucial step for EBV to evade the immune response and initiate the latency programs.

As most epigenetic modifications are reversible, manipulating this complex machinery could be critical in determining the outcome of the viral pathogenesis and the induction of tumorigenesis. For example, several studies in HIV patients have focused on the eradication of viral reservoirs by reactivating the latent virus using a wide range of epigenetic inhibitors followed by conventional antiviral therapy to neutralise the reactivated virus, otherwise known as “shock and kill” strategy [97]. The use of epigenetic inhibitors has been proposed for the treatment of HBV-mediated hepatocellular carcinoma (HCC). The regulatory HBV X protein (HBx) is a viral multifunctional molecule that enhances HBV replication, and it is implicated in the induction of HHC [98]. In particular, this protein is involved in several posttranslational modifications (PTM) by recruiting epigenetic modifiers, such as DNMTs and HDACs, and increasing the levels of H3K4me3 in target genes that are essential for cellular migration, invasion, and growth. Several epigenetic inhibitors targeting DNMTs (azacytidine, 5-AZA) and HDACs (trichostatin A, TSA) have been tested in the treatment of HBV-mediated HCC; however, they have had controversial results. While the administration of 5-AZA or TSA resulted in the reactivation of tumour suppressor genes, on the other hand, they increased HBV viral load and oncogene expression [99,100,101]. Similar approaches have been tested in the context of EBV-related lymphomagenesis, by targeting epigenetic modifiers as oncolytic therapy during both lytic and latent cycles. The lytic cycle reactivation by HDAC inhibitors, including trichostatin A, sodium butyrate, valproic acid, and SAHA, can lead to enhanced apoptosis in NPC and gastric carcinoma cells [102,103]. Comparably, HDACs inhibitors have been shown to stimulate apoptosis of the EBV+ B cells in Burkitt lymphomas in latency III stage by inducing the expression of p21^WAF1^ [104]. Similar approaches have been exploited for the clinical management of Hodgkin’s Lymphoma and DLBCL during latency II. The non-selective HDAC inhibitor Panobinostat exhibited a great anti-proliferative effect through activation of the caspase pathway in the HL cell line [105]. A recent clinical trial using Panobinostat, in combination with the cell cycle inhibitor Lenalidomide, has shown promising results in patients with relapsed or refractory HL [106]. The use of DNMT inhibitors azacitidine and decitabine as monotherapy or in combination with other epigenetic-modulating therapies is still under investigation, given the high toxicity of these drugs. However, chemo-resistant cells from refractory DLBCL patients became sensitive to chemotherapy after prolonged administration of low-dose decitabine, showing its potential application in the treatment of DLBCL [107].

An improved understanding of the modulation of the epigenetic machinery is revealing a tremendous array of novel targets; however, this potential proves to be a double-edged sword. To wit, in the context of HBV, the available epigenetic therapies lack specificity, which raises concerns about their cytotoxic side effects due to unintended global epigenetic modifications and complicates the goal of a therapeutic index within the acceptable toxicity levels. While the direct role of EBV viral proteins in the dysregulation of epigenetic modifiers has been broadly unveiled, it will be crucial to understand how other factors, such as the chronically inflamed environment, can affect the expression and function of these molecules. Revealing the molecular basis underlying these mechanisms could pave the way for new therapeutic approaches targeting epigenetic players and lead to major clinical breakthroughs in the future.

## Figures and Tables

**Figure 1 cancers-12-03037-f001:**
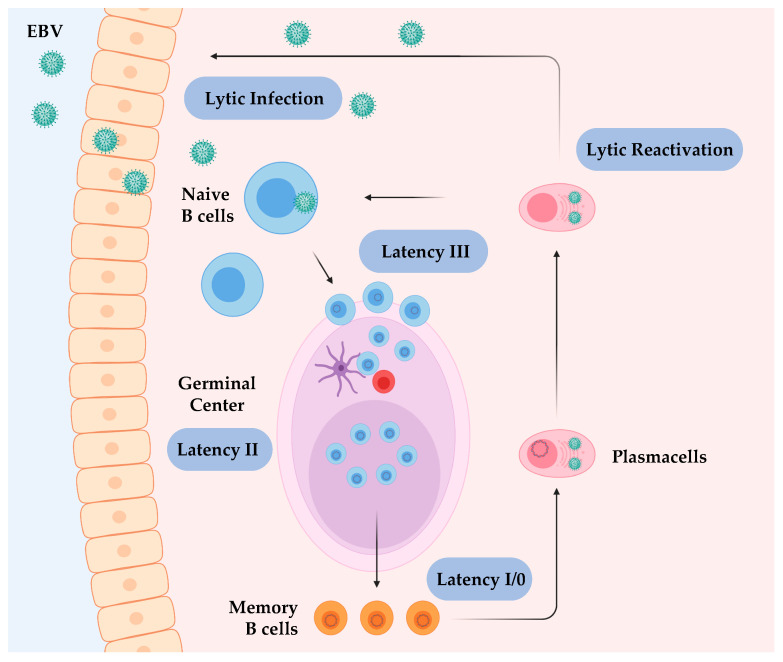
Epstein–Barr Virus (EBV) life cycle and latency stages. The viral life cycle includes at least five different stages (Lytic infection, Latency III, Latency II, Latency I/0, and Lytic reactivation), and four of them are associated with EBV diseases. EBV infects submucosal B cells, inducing viral gene transcript expression that establishes the latency III program. The infected cells pass through the lymph node germinal centre, proliferate and mature. During this stage, EBV induces gene expression of the Latency II program. Some latently infected memory B cells leave the germinal centre and persist (Latency I/0), whereas occasionally some infected memory cells evolve to plasma cells that shed newly assembled free virions into saliva (lytic reactivation). Credits: Created with BioRender.com.

**Figure 2 cancers-12-03037-f002:**
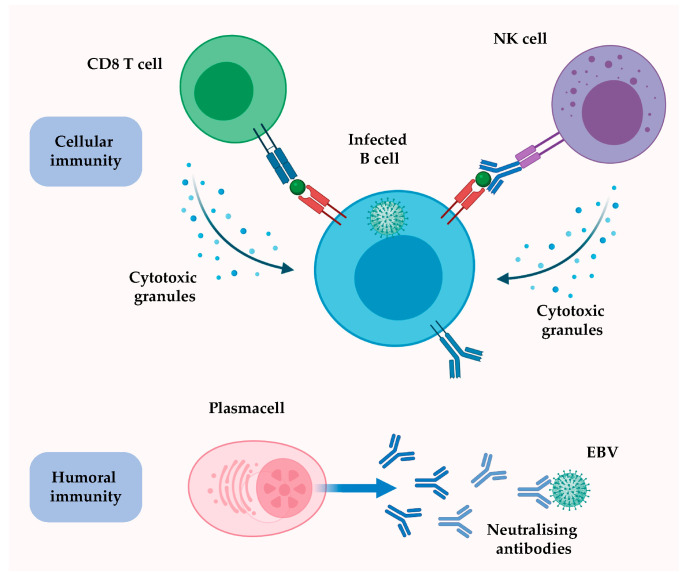
The targeting of EBV-infected cells and EBV virions by the adaptive immune system. EBV-infected B cells that display viral antigens on major histocompatibility (MHC) molecules are recognized by cytolytic T cells (CD8 T cells), which release cytotoxic granules (e.g., perforin and granzymes) and trigger apoptosis in infected cells. The binding of antibodies to glycoproteins at the surface of replicating cells enable their recognition and elimination by natural killer (NK) cells through antibody dependent-cellular cytotoxicity (ADCC). Humoral immunity prevents the spreading of the infection to other B cells of the host by targeting EBV virions through neutralising antibodies. Credits: Created with BioRender.com.

**Table 1 cancers-12-03037-t001:** Summary of EBV latent transcripts and their functions in epigenetic deregulation during B-cell lymphomagenesis.

Latent Transcripts	Latency Program	Functions	B Cell Lymphoma
EBNA2	III	Major EBV-encoded transcriptional activator, inducing gene transcription, such as cMyc.	Diffuse large B cell lymphomas (DLBCLs), Immunoblastic lymphomas
EBNA-LP	III	Acts as a co-transactivator of EBNA-2 expression by deregulating HDAC (HDAC4 and HDAC5) activities.	Diffuse large B cell lymphomas (DLBCLs), Immunoblastic lymphomas
EBNA3A/C	III	BCL2L11 promoter repression through CpG-methylation and recruiting PRC2 complex, H3K27me3 heterochromatic mark. Represses CDKN2A through recruiting CtBP, depositing H3K27me3. Transcriptional regulation through interacting with several HATs and HDACs. Inhibits CDKN2B transcriptions through induction of H3K27me3 heterochromatic mark.	Diffuse large B cell lymphomas (DLBCLs), Immunoblastic lymphomas
LMP1	III/II/I/0	Interacts with the DNA methyltransferase DNMT3B, regulating cellular apoptosis by elevating antiapoptotic Bcl2 expression Induces DNMT1, which is involved in the JNK-AP-1 pathway	Diffuse large B cell lymphomas (DLBCLs), Immunoblastic lymphoma, Hodgkin lymphoma, Burkitt lymphoma
LMP2A	III/II/I/0	Induces DNMT1, which is involved in the JNK-AP-1 pathway.	Diffuse large B cell lymphomas (DLBCLs), Immunoblastic lymphoma, Hodgkin lymphoma, Burkitt lymphoma

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
