# Peer review of "Epstein–Barr Virus Promotes B Cell Lymphomas by Manipulating the Host Epigenetic Machinery"

_cancers, 2020, doi:10.3390/cancers12103037_

Round 1
Reviewer 1 Report
The multifaceted functions of Epstein-Barr virus in controlling the host epigenetic machinery by Andrea Di Pietro is a review article focusing on EBV associated B-cell lymphomas in terms of epigenetic modifications. The review largely sticks to B-cell lymphomas and does not discuss the other EBV cancers (EBV associated gastric carcinoma and NPC) and their regulation by epigenetic factors. Perhaps the title can be modified to to indicate this focus.
Current reviews on this topic are available but this review is somewhat different in that it includes a discussion and thought provoking section on chronic inflammation in the context of EBV infection.
Most of the discussion focuses on Type II latency (DLBCLs) but I think more could be added to expand discussion on Type 1(Burkitt) and II latency (Hodgkin) lymphomas. These latency types are largely defined by promoter usage (Qp, Cp, Wp) and are dependent on epigenetic factors. A section could be added describing this.
Minor points.
Lines 42-44. The authors should include KSHV (in addition to HBV, HCV, EBV, and HPV which were mentioned) as an oncogeneic virus which contributes to epigentic changes.
Line 112. "passe" should be "pass."
References can be added in places. Lines 210-212. lines 229-230. line 240-242.
Author Response
I would like to thank the reviewer for taking the time to assess my manuscript and for his appreciation regarding the provoking section on the potential of chronic inflammation during EBV infection in (dys)regulating the expression and functions of epigenetic modifiers, which might represent an alternative way in contributing to EBV-related tumorigenesis.
Regarding the reviewer's comment:
"Most of the discussion focuses on Type II latency (DLBCLs) but I think more could be added to expand discussion on Type 1(Burkitt) and II latency (Hodgkin) lymphomas. These latency types are largely defined by promoter usage (Qp, Cp, Wp) and are dependent on epigenetic factors. A section could be added describing this."
A paragraph has been added in the Conclusion section, discussing the potential of epigenetic modifier inhibitors towards the listed lymphomas: DLBCL, Burkitt and Hodgkin. References have been added accordingly.
Lines 42-44. The authors should include KSHV (in addition to HBV, HCV, EBV, and HPV which were mentioned) as an oncogeneic virus which contributes to epigentic changes.
The KSHV has been included, together with the corresponding reference.
Line 112. "passe" should be "pass."
Now line 113, it has been corrected.
References can be added in places. Lines 210-212. lines 229-230. line 240-242.
Lines 210-212: Now lines 229-231, references have been added;
Lines 229-230: Now line 249, reference has been added;
Lines 240-242: Now line 261, reference has been added.
Kind Regards,
Andrea
Reviewer 2 Report
Overall, this is a well written interesting review describing the interplay between viral proteins and cellular chromatin remodeling machineries.
Some minor points are suggested
- The title may be modified to include how the cellular epigenetic machineries also regulate the virus life cycle.
- P2, Line 87 “EBV potential”-> “the potential of EBV”
- Line 142. describes that EBV is maintained in memory cells. Two possibilities according to Reference 34 are suggested. Because this is a very important issue in terms of EBV biology and pathogenesis. Further explanation and the relevant evidence would help readers’ understanding. It will be great to include recent studies in this regard.
Author Response
I would like to thank the reviewer for taking the time to assess my manuscript and its kind general comments on the manuscript.
Regarding the reviewer's comment:
The title may be modified to include how the cellular epigenetic machineries also regulate the virus life cycle.
The title has been modified by adding the words "in B cells", given that most of the review is focused on this infected cell population.
P2, Line 87 “EBV potential”-> “the potential of EBV”
This is now corrected.
Line 142. describes that EBV is maintained in memory cells. Two possibilities according to Reference 34 are suggested. Because this is a very important issue in terms of EBV biology and pathogenesis. Further explanation and the relevant evidence would help readers’ understanding. It will be great to include recent studies in this regard.
The section (now lines 147-159) has been restructured following the schematics of Figure 1, to help the reader in better understanding the flow of latency establishment. Now more detailed information on the establishment of EBV latency has been added, including specifics on the different EBV promoter engagement and regulation. References have been added accordingly.
Kind Regards,
Andrea